# How attitudes of state and community leaders regarding health equity and social determinants of health are associated with behavioral intentions to improve population health

**Nyna Williams**[1]*, **Emily Newton-Hoe**[2], **Martha Kovac**[1]

**1** Division of Health Research, Mathematica, Princeton, NJ, United States of America, **2** Department of Social and Behavioral Sciences, Harvard University, Cambridge, MA, United States of America

* nwilliams@mathematica-mpr.com

**Data Availability Statement:** The data (and survey instrument) are available as a public use file in the Robert Wood Johnson Foundation's Health and

## Abstract

A critical driver of population health, well-being, and equity is making health a shared value, or the belief that health is a priority for oneself and one's community. For people in positions of influence, attitudes regarding their organization's perspective may result in decisions that advance health equity or reinforce inequities. Yet despite the potential role of leaders in influencing health and well-being, little is known about their attitudes regarding their organization's perspective on what shapes health, or how their attitudes translate into actions that improve population health. The objectives of this study are twofold: (1) to understand leaders' attitudes with regard to their organization's perspective on population health and (2) to identify the attitudes associated with leaders' perspectives regarding their organization's intention to engage in behaviors that advance population health. We designed a survey instrument grounded in the Theory of Planned Behavior and fielded the survey to 18,367 state and community leaders across all 50 U.S. states, 325 U.S. cities, and nine sectors from fall 2020 until spring 2022. We received surveys from 5,450 leaders, yielding a 32 percent response rate. Survey respondents viewed social determinants of health and health equity as important factors influencing population health. Additionally, two attitudes were most associated with leaders' intentions to promote population health—"self-efficacy" (a leader's confidence in taking action to promote health for all) and "behaviors valued by important others" (how much certain groups want the organization to take action that promotes health for all). Efforts to improve population health through leaders should focus on supporting leaders' self-efficacy and on increasing leaders' perceptions regarding behaviors valued by important others. Governmental entities and funders can use these data to support leaders' efforts in improving population health, well-being, and equity.

Medical Care Archive at the Inter-university Consortium for Political and Social Research at the University of Michigan (ICPSR 38770; Measuring Mindset and Attitude Shifts in State and Community Leaders, United States, 2020-2022; https://www.icpsr.umich.edu/web/HMCA/studies/38770).

**Funding:** This research was funded by the Robert Wood Johnson Foundation (www.rwjf.org) under contracts #76607, #79367, and #80560 awarded to Mathematica (not to the authors, all of whom are employed by Mathematica). The funders provided guidance and feedback throughout the research on the study design and decision to publish; however, they were not directly involved in implementation of data collection and analysis or preparation of the manuscript.

**Competing interests:** The authors have declared that no competing interests exist.

# Introduction

The Robert Wood Johnson Foundation's (RWJF) Culture of Health Action Framework provides a blueprint for ensuring that everyone in America has a fair and just opportunity for health and well-being [1]. According to the framework, a critical driver of population health, well-being, and equity is making health a shared value, or the belief that health is a priority for oneself and one's community [1, 2]. Personal views about whether individual and collective health are valued and prioritized may drive people "to act in ways that improve and protect their own health and the health of others" [2, 3]. For people in positions of influence in local communities, businesses, and state-level organizations, attitudes regarding their organization's perspective may result in decisions that advance health equity or reinforce inequities. Yet despite the potential role of leaders in influencing health and well-being, little is known about their attitudes regarding their organization's perspective on what shapes health, or how their attitudes translate into actions that improve population health.

There is a need to fill this gap in the evidence base—specifically how state and community leaders view, understand, and talk about the influences of health and well-being in communities, particularly in regard to social determinants of health and health equity. To our knowledge, no other surveys have sampled a nationally representative group of diverse, cross-sectoral leaders and asked about their attitudes toward health and well-being or their attitudes regarding their organization's perspective on health and well-being. Most surveys have focused on the health attitudes of the general population, such as the National Opinion Survey on Health and Health Disparities [4], the National Survey of Health Attitudes [5], the American Health Values Survey [6], and COVID-19 and the Experiences of Populations at Greater Risk Survey [7]. Other surveys have sampled subsets of leaders including American mayors [8] and businesspeople [9]. Still others have focused on the attitudes of people providing clinical care, such as health system executives [10] or faculty at medical schools [11]. While these surveys have generally found that people are aware of social, economic, and racial health disparities, recognize the influence of social and physical factors on health, and support interventions that promote population health and health equity, they tend to overlook important predictors of behavior change such as self-efficacy and social norms. As a result, we developed the *What Shapes Health and Well-Being* survey to produce evidence on what leaders from all 50 U.S. states, 325 U.S. cities, and nine sectors think are important drivers of actions and policies that promote health and well-being in their respective cities and states across the United States.

The objectives of this study are twofold: (1) to understand leaders' attitudes with regard to their organization's perspective on population health and (2) to identify the attitudes associated with leaders' perspectives regarding their organization's intention to engage in behaviors that advance population health. Findings from this study will fill the current gaps in evidence base about what people in positions of influence think about social determinants of health and health equity, as well as their attitudes regarding their organization's perspective. Furthermore, capturing these baseline data is a first step toward shifting societal expectations around health and making health a shared value [12].

# Materials and methods

## Conceptual framework

We created a conceptual framework to inform the development of the *What Shapes Health and Well-Being* survey instrument and sample plan. First, we identified eight framework constructs for the survey instrument pertinent to promotion of population health. Second, we identified five topics for the survey instrument pertinent to understanding attitudes related to

**Table 1. Constructs, topics, and sectors in conceptual framework.**

| Constructs | Topics | Sectors |
|---|---|---|
| 1. Beliefs regarding root causes of health<br>2. Beliefs regarding behavioral drivers of outcomes<br>3. Perceived importance of outcomes<br>4. Beliefs regarding behaviors valued by important others<br>5. Motivation to comply with important others<br>6. Beliefs regarding factors that impact behaviors perceived to drive outcomes<br>7. Self-efficacy to engage in behaviors perceived to drive outcomes<br>8. Promotion of policies and resource allocation perceived to drive outcomes | 1. Social determinants of health<br>2. Equity<br>3. Community engagement<br>4. Systems alignment<br>5. Allocation of resources | 1. Health care systems<br>2. Public health and social services<br>3. Housing<br>4. Transportation<br>5.Recreation/open space<br>6. Physical safety<br>7. Employment<br>8. Education<br>9. Economic/community development<br>10. Cross-cutting |

health and well-being. Third, we identified nine sectors and a cross-cutting sector upon which we based our sample parameters. Table 1 shows the constructs, topics, and sectors.

We grounded our approach in behavioral theory. After comparing several theories or models of behavior change using standard criteria, we selected the theory of planned behavior (TPB) [13]. Table 2 shows the mapping of our framework constructs to the TPB constructs. As shown in Fig 1, the theory states that behavioral intention depends upon three predictor variables—attitude toward the behavior, subjective norms, and perceived behavioral control. Each predictor variable consists of two constructs (an "expectancy" construct and a "value construct). In addition, behavioral intention predicts behavior, which is also predicted by perceived behavioral control.

A technical expert panel (TEP) of researchers and thought leaders provided guidance throughout the design, data collection, and analysis. The TEP consisted of nine members representing grassroots organizations, local nonprofits, local health collaboratives, mindset and attitude experts, and sector-specific experts.

## Survey methods

The survey instrument focused on social determinants of health and equity—the topics deemed most pertinent for improvement of population health. To develop the instrument, we began by identifying survey questions through a literature review, and developed new

**Table 2. Framework constructs mapped to TPB constructs.**

| Framework construct | TPB construct |
|---|---|
| Beliefs regarding root causes of health | — |
| Beliefs regarding behavioral drivers of outcomes | Behavioral beliefs (attitude toward behavior **expectancy**) |
| Perceived importance of outcomes | Outcome evaluations (attitude toward behavior **value**) |
| Beliefs regarding behaviors valued by important others | Normative beliefs (subjective norms **expectancy**) |
| Motivation to comply with important others | Motivation to comply (subjective norms **value**) |
| Beliefs regarding factors that impact behaviors perceived to drive outcomes | Control beliefs (perceived behavioral control **expectancy**) |
| Self-efficacy to engage in behaviors perceived to drive outcomes | Power (perceived behavioral control **value**) |
| Intention to promote [promotion of] policies and resource allocation perceived to drive outcomes | Behavioral intention [behavior] |

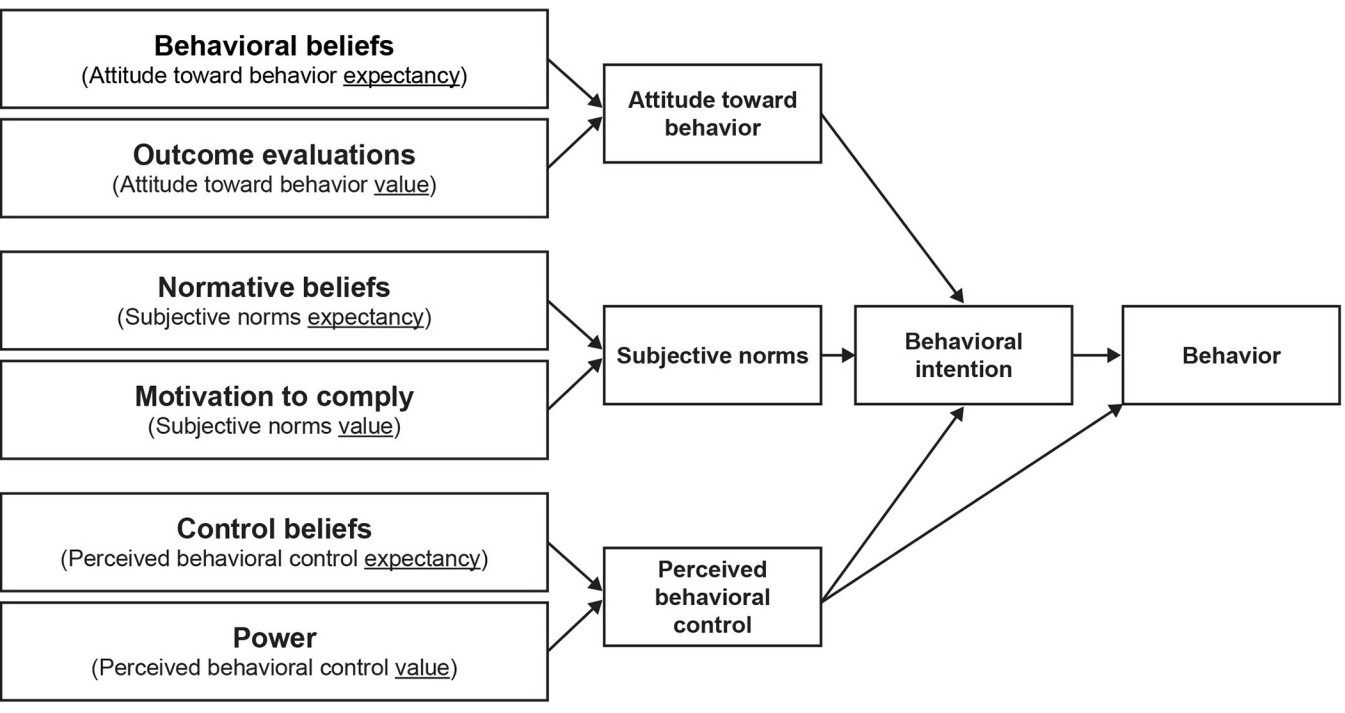

**Fig 1. Theory of planned behavior (TPB) model.**

questions as needed. We formatted survey questions as tables with lists of items in a grid list format with 5-point Likert scale response options (all scaled 1–5) and combined similar questions to reduce the cognitive burden. We conducted 18 pre-test interviews of the draft survey instrument to measure interview duration, test the item wording and flow, and gather feedback on the questionnaire. Changes were made based on respondent feedback. Table 3 presents the question language for each framework construct. All questions except Q1 ask for the leader's perspective regarding their organization.

Our survey instrument, contact materials, and passive consent process were reviewed and approved by the Western Institutional Review Board (WIRB) Copernicus Group (WCG) (#20202733). For our passive consent process, we provided respondents with information on the survey purpose, length, intended use, and confidentiality of the data; conveyed that participation in the survey was voluntary and that the study was being overseen by an institutional review board (IRB); and provided a phone number and email address for the IRB if they had questions or concerns. During the process of sample frame development, sample selection, survey fielding, and data processing and weighting, the research team had access to the names of individual participants. The data were stored on secure, encrypted, and password-protected network drives. Only the research team had access to the files with individual participant names.

The target population was organizational leaders in city and state government and leaders of for-profit and nonprofit organizations. We selected two, mutually exclusive samples of leaders. We drew one sample from state-level organizations or agencies in all 50 states, and we drew the other sample from city-level organizations and agencies in 325 cities across the U.S. Both samples consisted of leaders in a wide array of sectors and were nationally representative.

We fielded the survey in two phases—Phase 1 ran from October 9, 2020, to May 31, 2021 (with a sample of 13,193 leaders), and Phase 2 ran from September 28, 2021, to March 31, 2021

**Table 3. Question language for framework constructs.**

| Question number | Framework construct | Question language |
|---|---|---|
| 1 | Beliefs regarding root causes of health | Stem—Here is a list of some things that may affect people's health and well-being. How strong an effect does each have?<br>Grid list items—health care, community conditions, behavioral choices, genes<br>Response options—No effect (1) to Very strong effect (5) |
| 2 | Perceived importance of outcomes (SDOH) | Stem—How important is it to your organization that all residents have access to the following?<br>Grid list items—living wage job, healthy and affordable food, quality and affordable housing, quality K-12 education, quality and affordable childcare, safe neighborhood, clean air and water, affordable health insurance, quality and affordable health care, quality and affordable mental and behavioral health care, public transportation<br>Response options—Very unimportant (1) to Very important (5) |
| 3 | Motivation to comply with important others | Stem—How important is it to your organization to be responsive to each of the following groups?<br>Grid list items—residents, community and grassroots organizations, businesses, law enforcement, government leaders, health care organizations, social service organizations, public health organizations, organizations that provide funding<br>Response options—Very unimportant (1) to Very important (5) |
| 4 | Perceived importance of outcomes (equity) | Stem—How important is it to your organization that each of the following groups has the same opportunities to be healthy as other residents?<br>Grid list items—people of color, people living in poverty, people living in rural areas, people living with disabilities, immigrants<br>Response options—Very unimportant (1) to Very important (5) |
| 5 | Beliefs regarding behavioral drivers of outcomes | Stem—To what extent does your organization think these actions will increase all residents' opportunities to be healthy?<br>Grid list items—promoting program and policy initiatives that prioritize the well-being of residents from disadvantaged communities; promoting fair and just distribution of resources; partnering with residents throughout planning, development, and implementation; promoting better coordination and alignment across health care, social services, and public health systems; addressing inequalities faced by people of color in various sectors; promoting program and policy initiatives that prioritize the well-being of children and families<br>Response options: Not at all (1) to A great deal (5) |
| 6 | Self-efficacy to engage in behaviors perceived to drive outcomes | Stem—How confident is your organization in its ability to take these actions to increase all residents' opportunities to be healthy?<br>Grid list items—Same as Q5.<br>Response options—Not at all confident (1) to Extremely confident (5) |

(*Continued*)

**Table 3.** (Continued)

| Question number | Framework construct | Question language |
|---|---|---|
| 7 | Intention to promote [promotion of] policies and resource allocation perceived to drive outcomes | Stem—In the next two years, how likely is it that your organization will take these actions to increase all residents' opportunities to be healthy? Grid list items—Same as Q5. Response options—Very unlikely (1) to Very likely (5) |
| 8 | Beliefs regarding behaviors valued by important others | Stem—How much do each of the following groups want your organization to take action so that all residents have an opportunity to be healthy? Grid list items—Same as Q3. Response options—Not at all (1) to A great deal (5) |
| 9, 10, and 11 (reverse coded) | Beliefs regarding factors that impact behaviors perceived to drive outcomes | Q9 stem—To what extent are these items barriers to your organization taking action to increase all residents' opportunities to be healthy? Q9 grid list items—government policies or regulations, lack of resources, lack of political leader motivation, community beliefs about who should receive health and social services, community beliefs about the impacts of structural racism, lack of available health care services Q9 response options—Never a barrier (1) to Always a barrier (5) Q10 stem—To what extent are these items barriers to your organization promoting fair and just distribution of resources among residents? Q10 grid list items—lack of data on resource distribution, lack of political leader motivation, lack of coordination across sectors, community beliefs about who should receive health and social services, community beliefs about the impact of structural racism, lack of engagement between leaders and disadvantaged groups Q10 response options—Same as Q9 Q11 stem—To what extent are these items barriers to ongoing coordination and alignment across health care, social services, and public health systems? Q11 grid list items—lack of trust and communication across sectors; data privacy, systems, and measurement; lack of shared vision; difficulty changing established roles and relationships; lack of resources; lack of community involvement Q11 response options—Same as Q9 |

(with a sample of 5,174 leaders). We administered both survey phases via web, mail, and computer-assisted telephone interviewing. Each survey phase was fielded over a 22-week period, and each phase included sending emails, paper questionnaires, reminder letters, and telephone calls to encourage response.

We fielded the survey to 18,367 leaders and received survey responses from 5,450 leaders, achieving a response rate of 32 percent. Our respondents were somewhat older than the general adult population (37 percent 60 years of age or older), which is not surprising, given that they are leaders. Most had been working five or more years in their current job (58 percent). The gender split was fairly even (57 percent male) and the race/ethnicity mix was similar to that of the general adult population (76 percent non-Hispanic white). Most important, we found no nonresponse bias by gender or by race/ethnicity.

## Statistical methods

We conducted our analyses using the IBM SPSS Statistics Version 28 software's Complex Samples procedures, to account for the complex sample design and to apply the weighting for selection probability with nonresponse adjustments for the race and gender of the respondents and for the region, city population size, and city area deprivation index scale, and normalized to roughly equate each stratum.

We conducted a factor analysis for each framework construct question, confirming that each question loaded on a single factor. Next, we conducted a factor analysis with all of the framework construct questions, which showed that three questions loaded together on a single factor, and thus should be combined into a single question. Finally, we confirmed internal consistency reliability for each scale by calculating the Cronbach's α for each scale. The Cronbach's α for each scale was .91 or above, which was very high and meant that we could treat each question as a scale.

We calculated scores for each framework construct based on responses to the survey questions for that construct; we did this by calculating the mean for the grid list items for that question. Scores range from 1 to 5, with higher scores indicating more positive attitudes. For three constructs higher scores indicated greater perceived barriers and therefore were reverse-coded (see Q9–Q11 in Table 3).

We calculated scores for the TPB variables using an algorithm that combined the "expectancy" and "value" questions for the three TPB variables. We modified an established algorithm [14], because we did not have parallel grid list items for the "expectancy" and "value" questions for the three TPB variables, as required for the established algorithm. Scores ranged from -10 to +10, with higher scores indicating higher levels of the variable. The data (and survey instrument) are available as a public use file in the RWJF's Health and Medical Care Archive at the Inter-university Consortium for Political and Social Research at the University of Michigan (ICPSR 38770) [15].

## Results

We conducted three analyses—a descriptive analysis and two regression analyses. One regression analysis was based on the conceptual framework constructs, and one was based on the TPB variables; this provided two different ways to assess which attitudes predict behavioral intention. The framework analysis provides a more granular exploratory assessment that included all of the constructs, whereas the TPB assessment provides an assessment grounded in behavioral theory.

### Descriptive analysis

The aim of the descriptive analysis was to understand leaders' attitudes regarding population health. Survey respondents reported that the choices people make about their diet, exercise, and other behaviors most influence their health and that people's genetics/DNA have a lesser effect (means of 4.65 and 3.77, respectively, on a 1–5 scale; Q1c and Q1d). In general, they viewed social determinants of health and health equity as important factors influencing their community's health (means of 4.57 and 4.65, respectively, on a 1–5 scale; Q2 and Q4). Respondents also felt it was somewhat important to be responsive to certain groups (such as residents, businesses, government leaders, and community groups) in their community (mean of 4.29 on a 1–5 scale; Q3) and that these groups somewhat wanted their organization to take action to increase opportunities to be healthy (mean of 3.22 on a 1–5 scale; Q8). Finally, although respondents felt it was important to take action to increase opportunities to be healthy (mean of 4.19 on a 1 to 5 scale; Q5), many lacked confidence that their organization could or would

do anything (means of 3.24 and 3.73, respectively, on a 1–5 scale; Q6 and Q7). The barrier rated as most challenging was lack of resources, both to taking action to increase all residents' opportunities to be healthy and to ongoing coordination and alignment across health care, social services, and public health systems (means of 3.79 and 3.86, respectively, on a 1–5 scale; Q9 and Q11 grid list items).

## Framework analysis

The objectives of this analysis were (1) to assess relative values of the attitudes represented by the framework constructs and (2) to assess which of those attitudes were associated with intention to engage in behaviors perceived to advance population health.

**Comparisons among framework constructs.** Table 4 shows that the "self-efficacy" and "behaviors valued by important others" constructs had much lower scores than the other constructs (except one––"factors that impact behaviors"; see next paragraph). Therefore, there is the most room for improvement for those attitudes.

As mentioned above, the exception for the lowest relative scores is for the "factors that impact behaviors" construct. The questions associated with this construct were worded as perceived barriers, and so we reverse coded the responses (i.e., 1 = always a barrier and 5 = never a barrier). However, because those scores were close to the scale midpoint of 3 (i.e., "sometimes a barrier"), and because mean scores for the other constructs ranged from 3.22 to 4.65, scores for this construct would be the lowest either as originally scored (3.18 for perceived barriers) or reverse coded (2.82 for "factors that impact behaviors"). Paradoxically, the implication is that there is the most room for improvement both for increasing perceived barriers (from 3.18) and for increasing "factors that impact behaviors" (perceived barriers reverse coded; from 2.82). (Note that in the following regression analysis section, we also identify an unexpected finding for this construct).

**Regression analysis.** We conducted a regression analysis to assess which attitudes (framework constructs) predicted behavioral intention. For this analysis, we selected the backward elimination procedure, specifically because it is well-suited to exploratory research, effectively eliminating the least informative variables and retaining the strongest variables in the regression model. However, the backward elimination procedure is vulnerable to multicollinearity,

**Table 4. Comparisons among framework constructs.**

| Framework construct | Mean score (95 percent confidence interval) |
| --- | --- |
| Q1a (root causes of health/health care) | 4.45 (4.42, 4.49)$_c$ |
| Q1b (root causes of health/community conditions) | 4.29 (4.25, 4.33)$_d$ |
| Q1c (root causes of health/behavioral choices) | 4.65 (4.62, 4.68)$_a$ |
| Q1d (root causes of health/genes) | 3.77 (3.72, 3.81)$_f$ |
| Q5 (behavioral drivers of outcomes) | 4.19 (4.15, 4.23)$_e$ |
| Q2 (importance of outcomes/SDOH) | 4.57 (4.54, 4.60)$_b$ |
| Q4 (importance of outcomes/equity) | 4.65 (4.61, 4.69)$_a$ |
| Q8 (behaviors valued by important others) | 3.22 (3.17, 3.27)$_g$ |
| Q3 (motivation to comply with important others) | 4.29 (4.26, 4.33)$_d$ |
| Q9/Q10/Q11 (factors that impact behaviors) | 2.82 (2.78, 2.87)$_h$ |
| Q6 (self-efficacy) | 3.24 (3.19, 3.29)$_g$ |

Notes. Adjusted for the complex survey sample design and weighting. Scores were on a scale from 1 to 5. Means that share subscripts did not differ significantly at $p < 0.05$.

resulting in unstable regression results. Therefore, we conducted checks to assess multicollinearity and stability of the results.

- To assess multicollinearity, we examined correlations among all of the independent variables. Only 7 of 55 correlations, among the 11 independent variables, qualified as moderate (greater than or equal to +/- 0.4), and of those, only 1 qualified as strong (greater than or equal to +/- 0.6), the correlation between Q6 and Q8. This assessment suggests minimal multicollinearity.

- To assess stability of the results, we examined the univariate relationships between the independent and dependent variables using univariate linear regressions. Most important, all independent variables that were significant in the multivariate model were also significant in the univariate regressions. All additional variables except Q1c were significant in the univariate regressions; therefore, three variables (Q1d, Q2, and Q4) that are not significant in the multivariate model are significant in the univariate regressions. This assessment suggests reasonable stability.

We also conducted standard checks to assess the extent to which the data met the non-autocorrelation, noncollinearity, and normality assumptions of the linear regression procedure.

- We used the Durbin-Watson test to assess non-autocorrelation. The test statistic is 1.96, well within the recommended 1.5 to 2.5 range.

- We used the Tolerance and Variance Inflation Factor (VIF) tests to assess non-collinearity. The minimum Tolerance test statistics range from .53 to .88, all well above the recommended threshold of .10. The Maximum VIF test statistics range from 1.14 to 1.88, all well below the recommended threshold of 10.

- We assessed normality for all of the analytic variables with the Kolmogorov-Smirnov goodness-of-fit test. The test was significant for all variables at $p < 0.001$, indicating potential lack of normality. However, as recommended, we also examined skewness and the Q-Q plots for all of the variables. All variables were right-skewed, except Q9/Q10/Q11, which was left-skewed; however, skewness only exceeded the recommended +/- 2.0 threshold for Q1c, Q2, and Q4. Visual inspection of the plots for those variables confirmed that deviations from normality were small for Q1c and more substantive for Q2 and Q4. Linear regression is generally robust to violations of the normality assumption.

We regressed Q7 (behavioral intention) on Q1 as individual grid list items, Q2–Q6, Q8, and Q9, Q10, and Q11 as a combined variable (Q9/Q10/Q11), using backward stepwise elimination, with the criterion for inclusion set at 0.05, listwise deletion of missing data, and two-tailed tests of statistical significance. That is, we conducted a linear regression to predict behavioral intention (Q7) from the remaining framework constructs.

Table 5 shows that, after eliminating variables not significant in the first step and then eliminating variables not significant in the second step, all remaining variables were significant in the second step and no further backward elimination was required. The $R^2$ values (percentage of variance explained) at both steps were very good.

The following were the key findings:

- By far, the strongest effect was for "self-efficacy" (Q6)—that is, confidence in the organization's ability to take action—with a coefficient of 0.42.

- The effect for "behaviors valued by important others" (Q8)—that is, how much certain groups want the organization to take action—was also strong, with a coefficient of 0.32.

**Table 5. Standardized coefficients for framework constructs associated with behavioral intention.**

| Step 1: All variables ($N$ = 5,091; $R^2$ = 0.63) | |
|---|---|
| Q1a (root causes of health/health care) | -.05, $p < 0.001$ |
| Q1b (root causes of health/location) | .04, $p = 0.006$ |
| Q1c (root causes of health/choices) | -.03, $p = 0.058$ |
| Q1d (root causes of health/genes) | -.01, $p = 0.775$ |
| Q5 (behavioral drivers of outcomes) | .11, $p < 0.001$ |
| Q2 (importance of outcomes/SDOH) | -.01, $p = 0.827$ |
| Q4 (importance of outcomes/equity) | .02, $p = 0.134$ |
| Q8 (behaviors valued by important others) | .32, $p < 0.001$ |
| Q3 (motivation to comply with important others) | .07, $p < 0.001$ |
| Q9/Q10/Q11 (factors that impact behaviors) | -.11, $p < 0.001$ |
| Q6 (self-efficacy) | .42, $p < 0.001$ |
| Step 2: Variables significant in Step 1 ($N$ = 5,117; $R^2$ = 0.63) | |
| Q1a (root causes of health/health care) | -.06, $p < 0.001$ |
| Q1b (root causes of health/location) | .04, $p = 0.003$ |
| Q5 (behavioral drivers of outcomes) | .11, $p < 0.001$ |
| Q8 (behaviors valued by important others) | .32, $p < 0.001$ |
| Q3 (motivation to comply with important others) | .08, $p < 0.001$ |
| Q9/Q10/Q11 (factors that impact behaviors) | -.11, $p < 0.001$ |
| Q6 (self-efficacy) | .42, $p < 0.001$ |

Note. Adjusted for the complex survey sample design and weighting.

- The coefficients for the remaining constructs were all less than or equal to +/- 0.11. Although the effects for these constructs were statistically significant, they were weak.

- The negative effect for "factors that impact behaviors" (Q9/Q10/Q11), albeit a weak effect, was unexpected. These questions were worded as perceived barriers, and therefore, were reverse coded. The finding is that greater perceived barriers to taking action were associated with *greater* behavioral intention to take action. This finding was unexpected because theories of behavior change assume that behavioral facilitators increase behavior and behavioral barriers reduce behavior. Sensitivity tests confirmed that the weak positive association between perceived barriers and behavioral intention held up across sectors and economic and social ideologies.

## TPB analysis

The objectives of this analysis were (1) to assess relative values of "attitude toward the behavior" (favorability toward the behavior), "subjective norms" about the behavior (social pressure to take action), and "perceived behavioral control" (the organization's perceived ability to take action, given facilitators and barriers) and (2) to assess relative associations of "attitude toward the behavior", "subjective norms", and "perceived behavioral control" with intention to engage in behaviors perceived to advance population health.

**Comparisons among TPB variables.** Table 6 shows that the scores for "attitude toward the behavior" was the highest, followed by the scores for "subjective norms" (social pressure to take action), and then the scores for "perceived behavioral control" (the organization's perceived ability to take action, given facilitators and barriers). Therefore, there is the most room for improvement for "perceived behavioral control" and also a lot of room for improvement for "subjective norms".

**Table 6. Comparisons among TPB variables.**

| TPB variable | Mean score (95 percent confidence interval) |
|---|---|
| Attitude toward the behavior | 6.94 (6.79, 7.08)$_a$ |
| Subjective norms | 4.46 (4.33, 4.59)$_b$ |
| Perceived behavioral control | 0.64 (0.50, 0.79)$_c$ |

Notes. Adjusted for the complex survey sample design and weighting. Scores were on a scale from -10 to +10. Means that share subscripts did not differ significantly at $p < 0.05$.

**Regression analysis.** We conducted a regression analysis to assess the relative associations of the three TPB variables with behavioral intention. For this regression analysis, we selected the forced entry procedure in alignment with established protocols for the TPB. To the extent we have multicollinearity, the forced entry regression procedure could result in unstable regression results. Therefore, we conducted checks to assess multicollinearity and stability of the results.

- To assess multicollinearity, we examined correlations among the three independent variables. Two of the 3 correlations qualified as moderate (greater than or equal to +/- 0.4), but none of those qualified as strong (greater than or equal to +/- 0.6). This assessment suggests the potential for multicollinearity.

- To assess stability of the results, we examined the univariate relationships between the independent and dependent variables using univariate linear regressions. All independent variables are significant in both the univariate model and the multivariate model. This assessment suggests reasonable stability.

We also conducted standard checks to assess the extent to which the data met the non-autocorrelation, non-collinearity, and normality assumptions of the linear regression procedure.

- We used the Durbin-Watson test to assess non-autocorrelation. The test statistic is 1.95, well within the recommended 1.5 to 2.5 range.

- We used the Tolerance and VIF tests to assess noncollinearity. The minimum Tolerance test statistics range from .55 to .73, all well above the recommended threshold of .10. The Maximum VIF test statistics range from 1.37 to 1.83, all well below the recommended threshold of 10.

- We assessed normality for all of the analytic variables with the Kolmogorov-Smirnov goodness-of-fit test. The test was significant for all variables at $p < 0.001$, indicating potential lack of normality. However, as recommended, we also examined skewness and the Q-Q plots for all of the variables. Q7 (the dependent variable), "attitude toward the behavior", "subjective norms", and "perceived behavioral control" were all right-skewed, but skewness did not exceed the recommended +/- 2.0 threshold for any of the variables. Visual inspection further confirmed that deviations from normality were small for all analytic variables. Linear regression is generally robust to violations of the normality assumption.

We regressed Q7 (behavioral intention) on "attitude toward the behavior", "subjective norms", and "perceived behavioral control", with listwise deletion of missing data, and two-tailed tests of statistical significance. That is, we conducted a linear regression to predict behavioral intention (Q7) from the three TPB variables.

Table 7 shows that the $R^2$ value (percent of variance explained) is very good.

The following are the key findings:

**Table 7. Standardized coefficients for TPB variables associated with behavioral intention.**

| ($N = 5{,}112$; $R^2 = 0.54$) | |
| --- | --- |
| Attitude toward the behavior | .05, $p = 0.016$ |
| Subjective norms | .27, $p < 0.001$ |
| Perceived behavioral control | .54, $p < 0.001$ |

Note. Adjusted for the complex survey sample design and weighting.

- By far, the strongest effect was for "perceived behavioral control"—that is, the organization's perceived ability to take action, given facilitators and barriers—with a coefficient of 0.54.

- The effect for "subjective norms"—social pressure to act—was also strong, with a coefficient of 0.27.

- The coefficient for "attitude toward the behavior" was only 0.05. Although the effect was statistically significant, it was very weak.

## Discussion

We conducted this study to understand how state and community leaders' view social determinants of health and health equity from the perspective of their organization, and the extent to which their attitudes are associated with their organization's intention to engage in behaviors that advance population health. This study contributes to the growing evidence on health attitudes throughout the U.S. [4–7], while also providing new insight by focusing more narrowly on the attitudes of people in positions of influence in local communities, businesses, and state-level organizations. Consistent with other surveys of leaders [8], we found that leaders in our sample believe that their organizations want to engage in behaviors that improve population health. Leaders reported positive attitudes related to their organizations engaging meaningfully with residents, collaborating across sectors, addressing inequities in the health sector, promoting equitable resource allocation, and investing in programs that prioritize the well-being of disadvantaged communities, children, and families. At the same time, leaders cited various barriers that inhibit their organization's ability to take these actions. In contrast to other surveys of leaders—which qualitatively describe survey results [8–11]—we also conducted two complementary regression analyses to predict a leader's organization's intention to engage in behaviors that promote population health, a framework regression analysis that provides granular assessment of all of our constructs and a TPB assessment that is grounded in behavioral theory.

Findings from the framework regression analysis showed that the strongest predictor of behavioral intention is the leader's confidence in their organization's ability to take action ("self-efficacy") to improve population health. The second strongest predictor was the leader's belief regarding how much certain groups (such as residents, businesses, government leaders, and community groups) want the organization to take action ("behaviors valued by important others") to improve population health. In addition, the greatest room for improvement (that is, the lowest relative scores) is for these same constructs ("self-efficacy" and "behaviors valued by important others").

Findings from the TPB regression analysis showed that all three TPB variables—"attitude toward the behavior", "subjective norms", and "perceived behavioral control"—predict a leader's intention to engage in behaviors that increase residents' opportunities to be healthy. Similar to the framework analysis, the TPB regression analysis showed that the strongest predictor

of behavioral intention was the organization's perceived ability to take action, given facilitators and barriers ("perceived behavioral control", for which one of the two component constructs is "self-efficacy"). The next strongest predictor was social pressure to take action ("subjective norms", for which one of the two component constructs is "behaviors valued by important others"). In addition, the greatest room for improvement (that is, the lowest relative scores) is in increasing leaders' views in their organization's perceived ability to take pertinent action, given facilitators and barriers ("perceived behavioral control"). There is also room for improvement (that is, the next lowest relative scores) in increasing leaders' views of social pressure to take pertinent action ("subjective norms").

Given these findings, efforts to improve population health via leaders should focus on supporting leaders' "self-efficacy"––that is, confidence in taking actions that promote health for all. For example, governmental entities and funders could provide resources and tools that help organizations act to promote equity, cross-sector collaboration, and meaningful community engagement. Leaders could also support capacity-building efforts to help organizations engage in cross-sector collaborations and partner authentically with residents most impacted by health inequities. Finally, they could share data that illustrates how initiatives focused on equity, cross-sector collaboration, and meaningful community engagement improve the health of all residents.

Given these findings, efforts to improve population health should also focus on increasing leader's perceptions regarding "behaviors valued by important others"––that is, beliefs regarding how much certain groups want their organization to take action that promotes health for all. For example, governmental entities and funders could share information on the importance of effectively engaging with residents, businesses, government leaders, and community groups to better understand resident needs. They could also offer strategies on how to be responsive to these groups to increase residents' opportunities to be healthy. Finally, they could share information showing that these other groups believe it is important to take action to increase opportunities to be healthy.

The finding from the framework regression analysis that greater perceived barriers to action were associated with *greater* behavioral intention was unexpected and held up across sectors and economic and social ideologies. Future research should examine how respondents are interpreting perceived barriers and disentangle the relationships among perceived barriers, self-efficacy, behavioral intention, and behavior.

Some of that research is already underway by our team for the current study. An initial literature scan showed that whereas self-efficacy is a significantly strong predictor of both behavioral intention and behavior in analyses using a variety of behavior change models, perceived behavioral control is often a weak or nonsignificant predictor in TPB analyses. Furthermore, Tomczyk et al. [16] found that whereas perceived behavioral control had a nonsignificant effect, splitting perceived behavioral control into self-efficacy and factors that impact behaviors—the two component constructs of perceived behavioral control—yielded a strong positive effect for self-efficacy and a nonsignificant negative effect for factors that impact behaviors, for both behavioral intention and behavior. We are currently conducting a more robust literature review of previous research examining those relationships. We are also engaging a vendor to field an omnibus survey with a sample of respondents in leadership positions to explore those relationships, including testing the hypothesis that perceived barriers increase behavioral intentions by increasing the motivation to overcome the barriers. Finally, we are also conducting qualitative research to explore those relationships.

In addition, in a future wave of this survey, we could assess the extent to which leader attitudes assessed in the current baseline survey predict future behavior. We could also test the hypothesis that whereas greater perceived barriers are associated with greater behavioral

*intention*, perhaps by increasing motivation to overcome the barriers, perceived barriers may inhibit *actual* behavior.

In conclusion, we believe this is the first survey that has documented the health attitudes of a nationally representative sample of leaders. Our survey highlights that leaders understand the importance of social determinants of health and health equity; meaningfully engaging with community members to improve health and well-being; collaborating across sectors to affect change; and equitably allocating resources so that all people have the opportunity to be healthy. Our survey also finds that two attitudes are most associated with leaders' intentions to promote population health—"self-efficacy" (confidence in taking action) and "behaviors valued by important others" (how much certain groups want the organization to take action).

S1 Checklist provides the STROBE checklist confirming inclusion of all items recommended for reports of observational studies.

## Supporting information

**S1 Checklist. STROBE checklist.**
(PDF)

## Acknowledgments

We thank Carolyn Miller, Maryam Khojasteh, and Oktawia Wojcik from the Robert Wood Johnson Foundation for their guidance; former Mathematica colleagues Jared Coopersmith and Shauna Robinson for their contributions to design and analysis of the survey; and Community Catalyst for their support constructing the sample frame.

## Author Contributions

**Conceptualization:** Nyna Williams, Emily Newton-Hoe, Martha Kovac.

**Data curation:** Martha Kovac.

**Formal analysis:** Nyna Williams, Emily Newton-Hoe.

**Funding acquisition:** Nyna Williams, Emily Newton-Hoe, Martha Kovac.

**Investigation:** Martha Kovac.

**Methodology:** Nyna Williams, Emily Newton-Hoe, Martha Kovac.

**Project administration:** Nyna Williams, Martha Kovac.

**Software:** Nyna Williams.

**Supervision:** Nyna Williams, Martha Kovac.

**Visualization:** Nyna Williams, Emily Newton-Hoe.

**Writing – original draft:** Nyna Williams, Emily Newton-Hoe.

**Writing – review & editing:** Nyna Williams, Emily Newton-Hoe, Martha Kovac.

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
