## [Decision Letter · Decision Letter 0]

17 Mar 2024

How attitudes of state and community leaders regarding health equity and social determinants of health are associated with intentions to improve population health

PONE-D-23-06878

Dear Dr. Williams,

We’re pleased to inform you that your manuscript has been judged scientifically suitable for publication and will be formally accepted for publication once it meets all outstanding technical requirements.

Kind regards,

Professor Khalid Mehmood

Academic Editor

PLOS ONE

Additional Editor Comments (optional):

We are pleased to inform you that your manuscript, " How attitudes of state and community leaders regarding health equity and social determinants of health are associated with intentions to improve population health," has been accepted for publication in the PLOS One. Congratulations

!

In closing, thank you very much again for working with us to bring this work to fruition. We hope that as a member of the PLOS One community, you will continue to help the journal as a reviewer, use our articles (including your own) as a source of citation in your future work, and continue to submit your best work on health equity to PLOS One. In particular, please do try to cite the most recent of the published articles in PLOS One.

Congratulations again, and thank you for your contribution to PLOS One !

Reviewers' comments:

Reviewer's Responses to Questions

**Comments to the Author**

1. Is the manuscript technically sound, and do the data support the conclusions?

Reviewer #1: Yes

Reviewer #2: Yes

2. Has the statistical analysis been performed appropriately and rigorously? 

Reviewer #1: I Don't Know

Reviewer #2: Yes

3. Have the authors made all data underlying the findings in their manuscript fully available?

Reviewer #1: Yes

Reviewer #2: Yes

4. Is the manuscript presented in an intelligible fashion and written in standard English?

Reviewer #1: Yes

Reviewer #2: Yes

5. Review Comments to the Author

Reviewer #1: The statistical analysis is a bit more advanced than my level, but was logical to the extent to which I was able to follow it, and fully explanatory. The authors have explained all the various ways in which they ensured rigour in their analysis.

Reviewer #2: The manuscript by Williams et al. focused on analyzing organizations' perspectives with regards to attitudes of leaders and people in positions of influence towards health equity. The manuscript is well written, considers broad perspective views through their questionnaire and large cohort group. The regression analysis are well performed and explained. As a result, the manuscript seems fine for acceptance in current form.

6. PLOS authors have the option to publish the peer review history of their article (what does this mean?). If published, this will include your full peer review and any attached files.

Reviewer #1: No

Reviewer #2: No

---

## [Editor Report · Acceptance letter]

26 Mar 2024

PONE-D-23-06878 

PLOS ONE

Dear Dr. Williams, 

I'm pleased to inform you that your manuscript has been deemed suitable for publication in PLOS ONE. Congratulations! Your manuscript is now being handed over to our production team.

Kind regards, 

on behalf of

Prof. Dr. Khalid Mehmood 

Academic Editor

PLOS ONE